Reinstatement of Phascolosoma (Phascolosoma) varians Keferstein, 1865 (Sipuncula: Phascolosomatidae) based on morphological and molecular data

Silva-Morales Itzahi itzahi.silva@estudianteposgrado.ecosur.mx
Departamento de Sistemática y Ecología Acuática, El Colegio de la Frontera Sur , Chetumal, Quintana Roo , Mexico
Magalhães Wagner
Electronic publication date: 2020 Oct 27
Publication date: 2020
Volume: 8
Electronic Location ID: e10238
Received 2020 Aug 10; Accepted 2020 Oct 4
Copyright: © 2020 Silva-Morales
Copyright year: 2020
Copyright holder: Silva-Morales
License: This is an open access article distributed under the terms of the Creative Commons Attribution License, which permits unrestricted use, distribution, reproduction and adaptation in any medium and for any purpose provided that it is properly attributed. For attribution, the original author(s), title, publication source (PeerJ) and either DOI or URL of the article must be cited.
License URL: https://creativecommons.org/licenses/by/4.0/

Keywords: Taxonomy, Barcoding, COI, Cryptic species, Greater Caribbean

Funding: CONACyT 951069 Itzahi Silva-Morales was supported by a scholarship from CONACyT (951069). The funders had no role in study design, data collection and analysis, decision to publish, or preparation of the manuscript.

==============================
Phascolosoma (P.) varians, a sipunculan species known from the Greater Caribbean, was designated as a synonym of Phascolosoma (P.) nigrescens, which was originally described from Fiji. Their synonymy was primarily based upon an interpretation that these two species were morphologically indistinguishable. After its designation as a synonym, no further detailed analyses of morphological or molecular characteristics were performed to corroborate the assumed widespread distribution of Phascolosoma (P.) nigrescens. In this study, Phascolosoma (P.) varians is redescribed, and notable differences between this species and its proposed senior synonym are presented. These two species differ in the shape of their hooks, the spatial attachment of nephridia to the body wall, and the morphology of the contractile vessel. Additionally, there is high genetic divergence between nucleotide sequences within their respective cytochrome c oxidase subunit 1 (COI) genes, which supports the morphological data. Herein, the synonymy of Phascolosoma (P.) varians with Phascolosoma (P.) nigrescens is rejected due to morphological and molecular differences. Furthermore, the assumed widespread distribution of Phascolosoma (P.) nigrescens is still considered as questionable.

Introduction

The phylum Sipuncula comprised of 320 species as recorded by Stephen & Edmonds (1972), but after numerous revisions over a period of approximately 20 years by Edward and Norma Cutler, the total number was reduced to 149 valid species (Cutler, 1994). The reduced number of sipunculan taxa may call into question previous records, lead to possible taxonomic errors among field investigators, and suggest that most, if not all, species have been accounted for. However, after 1994, 13 new species have been described (Kawauchi & Rice, 2009; Hylleberg, 2013; Saiz et al., 2015; Silva-Morales et al., 2019). Importantly, in the Greater Caribbean region alone, 38 valid species have been recorded, of which 40% correspond to species with type localities outside of the Greater Caribbean, with seven of those species belonging to Phascolosomatidae (Quiroz-Ruiz & Londoño-Mesa, 2015).

Reduction in the number of species by Cutler (1994) was performed by proposing extensive lists of synonyms, and when utilized by subsequent investigators, those lists have most likely led to a number of incorrect species identifications. For example, in the Greater Caribbean, P. (P.) nigrescens has been reported by Cutler & Schulze (2004) from Barbados, Schulze & Rice (2004) from Belize, Schulze (2005) from Panama, and Frontana-Uribe, Hermoso-Salazar & Solís-Weiss (2018) from the Mexican Caribbean. These records may reflect the fact that Cutler (1994) considered the distribution of P. (P.) nigrescens as widespread and circumtropical, found generally between 30° N and 30° S in shallow waters of the Indian, Pacific, and Atlantic oceans.

The extensive synonymization approach by Cutler was proposed in part by an assumption that many sipunculan species have wide geographic distributions. Those with wide distributions were thought to be possible due to the high dispersal capability of species with teleplanic pelagosphera larvae, which were inferred to remain in the water column for up to 6 months base upon laboratory experiments (Rice, 1976). Phascolosoma (Phascolosoma) varians has a Category 4 developmental pattern (Rice, 1970): Indirect development with two pelagic larval stages, trochophore and planktotrophic pelagosphera. Planktotrophic pelagosphera larvae can be either short-lived (weeks) forms or larger long-lived (months) teleplanic larvae, as in P. (P.) varians (Boyle & Rice, 2014). The teleplanic larvae of the Phascolosomatidae are characterized by having cuticular papillae (Scheltema & Rice, 1990).

However, recent molecular analyses revealed potential taxonomic problems at the species level, where small morphological differences also were shown to correspond with distinct species, thus rejecting previously assumed wide distributions for some of those species (Staton & Rice, 1999; Kawauchi & Giribet, 2010; Schulze et al., 2012; Kawauchi & Giribet, 2014; Johnson et al., 2016; Silva-Morales et al., 2019). Because of this, Kawauchi, Sharma & Giribet (2012) proposed to clarify the taxonomic status of each species through a meticulous case-by-case analysis, integrating molecular data while considering that some species names that are currently hidden under synonyms deserve to be restored. In regards to molecular data, the mitochondrial cytochrome oxidase c subunit I (COI) an efficient identification tool for metazoan species, making it the core fragment for DNA barcoding (Hebert, Ratnasingham & De Waard, 2003).

Keferstein (1865) described Phascolosoma (Phascolosoma) varians from St. Thomas, West Indies, and P. (P.) nigrescens from Fiji. Cutler & Cutler (1983, 1990) reviewed the subgenus Phascolosoma (Phascolosoma), and redesignated 13 species, previously described and recorded from all around the world, as synonyms of P. (P.) nigrescens. The identification key by Cutler (1994) indicates that the diagnostic characters of P. (P.) nigrescens include the following: a distinct clear streak in particular hooks with observable swelling in the middle of vertical and horizontal portions. Phascolosoma (P.) varians was one of multiple species included as a junior synonym of P. (P.) nigrescens.

Herein, a detailed redescription of P. (P.) varians based upon topotypic specimens and additional material from other Caribbean localities is provided. Furthermore, P. (P.) varians is reinstated due to both morphological and molecular differences that distinguish it from its proposed senior synonym, P. (P.) nigrescens.

Materials and Methods

Specimens from the collections of the Marine Invertebrate Museum (UMML), Rosenstiel School of Marine and Atmospheric Science, University of Miami; Invertebrate Collections of the Florida Museum of Natural History (UF), University of Florida; and the Reference Collection of Benthos (ECOSUR) of El Colegio de la Frontera Sur, Chetumal, Mexico were reviewed.

Redescription of the species was primarily based upon a topotypic specimen, but additional materials from other Caribbean localities were also assessed for species-specific variations. Standardized descriptions included external and internal anatomy. The descriptions of hooks and papillae followed the terminology proposed by Cutler (1994). To measure the angle between the primary tooth and the hook, a line X was drawn perpendicular to the base through the most anterior part of the concave side, and a line Y was drawn from the tip until intersect X in the middle of the point (see Cutler, 1994:161–162, fig. 44A).

Hooks and papillae were extracted with fine forceps for examination under an Olympus CH30 compound light microscope. Hooks were excised from three different regions (proximal, median and distal) of the ringed area of the introvert. Papillae were described from three different regions (anterior, median and posterior) of the trunk, and also from the proximal introvert. Furthermore, these structures were examined using SEM to achieve a more detailed examination. For SEM preparation, the complete introvert was dehydrated through a series of increasing concentrations of hexamethyldisilazane (HMDS). Once air-dried, the introvert was mounted on an aluminum stub and coated with gold for observation with a JEOL JSM-6010Plus-LA scanning electron microscope at the Scanning Electron Microscopy Laboratory (LMEB), ECOSUR-Chetumal. Digital photographs of selected internal and external features were obtained with a Canon X6 digital camera mounted on a Leica MZ75 dissecting stereomicroscope. All images were rendered from a series of optical focal planes with HeliconFocus v6.7.1 (HeliconSoft Limited, 2007) to improve the depth of field for each specimen or set of specimens that were photographed.

For molecular analyses, eight COI sequences with an alignment length of 541 bp, from specimens identified as Phascolosoma (P.) nigrescens were retrieved from GenBank. One of them from Barbados (DQ300139), two from Florida (DQ300142, AY161122), one from Broome, Australia (DQ300143), two from New Caledonia (JN865121, JN865122), one from Israel (DQ300140), and another from South Africa (DQ300141). Also, a COI sequence from Phascolosoma (P.) granulatum Leuckart, 1828 (DQ300138) and two sequences from Phascolosoma (P.) agassizii Keferstein, 1866 (JQ904338, JQ904337) were included for comparison.

All sequences were aligned using the ClustalW method. Selection of the best model of substitution was determined according to the lowest Bayesian Information Criterion scores (BIC). From the BIC results, the Tamura 3-parameter (Tamura, 1992) model with a discrete Gamma distribution (+G) with five categories, assuming a fraction of sites is evolutionarily invariable (+I), was selected to construct a tree by maximum likelihood analysis. The Kimura 2-parameter model (Kimura, 1980) was used to estimate the average evolutionary divergence over sequence pairs within and between species. All analyses were carried out with Mega 7 (Kumar, Stecher & Tamura, 2016).

Results

Systematics

Family Phascolosomatidae Stephen & Edmonds, 1972

Phascolosoma Leuckart, 1828

Phascolosoma (Phascolosoma) Leuckart, 1828

Type species. Phascolosoma granulatum Leuckart, 1828

Diagnosis. Body wall muscles separated into distinct bands. Spindle muscle attached posteriorly; introvert hooks without accessory spinelets (after Cutler, 1994).

Phascolosoma (Phascolosoma) variansKeferstein, 1865 reinstated

Phascolosoma varians Keferstein, 1865: 424–426, pl. 32, Fig. 22. De Quatrefages, 1865: 623. Wesenberg-Lund, 1954: 7–8. Rice & Macintyre, 1982: 314.

Phymosoma varians: Selenka, 1883: 69–70, pl. 9, figs. 124–127. Shipley, 1890: 1–24, fig. pls. 1–4, figs. 1–32.

Physcosoma varians Ten Broeke, 1925: 85.

Phascolosoma (Phascolosoma) varians: Stephen & Edmonds, 1972: 327–328, fig. 39I.

Phascolosoma nigrescens Cutler, 1994 (partim): 179–181; Cutler & Schulze, 2004: 226; Schulze, 2005: 526; Frontana-Uribe, Hermoso-Salazar & Solís-Weiss, 2018:174, fig. 5a-b; (non Keferstein, 1865).

Material examined

USA, Florida. UMML 26.5, 2 specimens, Bear Cut, Key Biscayne, 25°43′54.33″N, 80°09′26.16″W, May 7, 1961, coll. P. Robertson. UMML s/n, 1 specimen. Margot Fish Shoal, Dade Co, Apr 5, 1966, coll. G. Hendrix, bored in coral rubble. UF 292, 1 specimen, N of St. Petersburg, 28°27′33.12″N, 84°16′18.48″W, 30 m, Mar 13, 2011, colls. G. Paulay, N. Evans, F. Michonneau, C. Thacker, R. Williams, A. Baeza. UF 293, 1 specimen, N of St. Petersburg, 28°27′33.12″N, 84°16′18.48″W, hard ground with sponges, 30 m, Mar 13, 2011, coll. G. Paulay. UF 324, 1 specimen, St. Petersburg, 28°33′24.12″N, 84°16′28.20″W, hard bottom, sponge reef, 27 m, May 24, 2012, coll. J. Slapcinsky. UF 325, 1 specimen, St. Petersburg, 28°39′03.96″N, 84°23′03.84″W, 26–30 m, hard bottom, sponge reef, May 25, 2012, colls. G. Paulay, N. Evans, F. Michonneau. Mexico, Mexican Caribbean, Isla Contoy. ECOSUR-S62, 3 specimens, Morro Norte, 21°28′32.79″N, 86°47′30.13″W, coralline rock, 2.5 m, Feb 26, 2008, colls. S. Salazar-Vallejo, L. Carrera-Parra. ECOSUR-S63, 2 specimens. Punta Sur, 21°27′37.10″N, 86°47′04.60″W, coralline rock, 1.5 m, Mar 2, 2001, colls. S. Salazar-Vallejo, L. Carrera-Parra. ECOSUR-S69, 1 specimen, Punta Sur, 21°27′37.10″N, 86°47′04.60″W, coralline rock, 1.5 m, Feb 28, 2001, colls. S. Salazar-Vallejo, L. Carrera-Parra. ECOSUR-S70, 1 specimen, Ixlache reef, 21°26′02.52″N, 86°46′56.16″W, coralline rock, 2 m, Feb 25, 2008, colls. S. Salazar-Vallejo, L. Carrera-Parra. Cancun, Punta Nizuc. ECOSUR-S87, 1 specimen, 21°01′41.92″N, 86°46′45.72″W, coralline rock, 2.6 m, Aug 31, 1997, colls. S. Salazar-Vallejo, L. Carrera-Parra, M. Ruiz-Zárate. ECOSUR-S88, 2 specimens, 21°01′17.06″N, 86°46′45.95″W, coralline rock, 4 m, Sep 1, 1997, colls. S. Salazar-Vallejo, L. Carrera-Parra, M. Ruiz-Zárate. ECOSUR-S89, 3 specimens, 21°01′17.06″N, 86°46′45.95″W, coralline rock, 4 m, Sep 1, 1997, colls. S. Salazar-Vallejo, L. Carrera-Parra, M. Ruiz-Zárate. ECOSUR-S90, 2 specimens, 21°01′17.06″N, 86°46′45.95″W, coralline rock, 4 m, Sep 1, 1997, colls. S. Salazar-Vallejo, L. Carrera-Parra, M. Ruiz-Zárate. Playa del Carmen. ECOSUR-S86, 4 specimens, Navega pier, 20°37′12.07″N, 87°04′26.63"W, fouling, 1 m, Aug 23, 2003, coll. M. Tovar-Hernández. Cozumel. ECOSUR-S64, 14 specimens, Playa Azul, 20°32′51.98″N, 86°55′46.45″W, coralline rock, 1 m, Mar 25, 2001, colls. S. Salazar-Vallejo, L. Carrera-Parra. ECOSUR-S65, 1 specimen, in front to SEDENA, 20°31′00.61″N, 86°56′45.52″W, coralline rock, 1.5 m, Mar 24, 2001, colls. S. Salazar-Vallejo, M. Londoño-Mesa. Tulum. ECOSUR-S66, 3 specimens, Playa Aventuras, 20°21′47.20″N, 87°19′53.10″W, coralline rock, 1.5 m, Feb 28, 1999, colls. S. Salazar-Vallejo, J. Bastida-Zavala. ECOSUR-S91, 1 specimen, Playa Aventuras, 20°21′47.20″N, 87°19′53.10″W, coralline rock, 1.5 m, Feb 18, 2001, colls. S. Salazar-Vallejo, L. Carrera-Parra. ECOSUR-S92, 4 specimens, Ana y José beach, 20°09′24.22″N, 87°27′13.74″W, coralline rock, 1 m, Feb 11, 2001, colls. S. Salazar-Vallejo, J. Bastida-Zavala, J. Ruiz-Ramírez. ECOSUR-S96, 3 specimens, Punta Piedra, 20°10′38.94″N, 87°26′42.28″W, coralline rock, 1.5 m, Feb 11, 2001, colls. S. Salazar-Vallejo, L. Carrera-Parra, M. Tovar-Hernández. Mahahual. ECOSUR-S67, 1 specimen, 25 m off coast, 18°43′27.09″ N, 87°42′3.64″W, reef lagoon, rocky substrate with sediment, 0.75 m, Oct 1, 1996, colls. S. Salazar-Vallejo, L. Carrera-Parra. ECOSUR-S71, 1 specimen, in sponge, Jul 21, 1998. ECOSUR-S72, 2 specimens, 50 m off coast, 18°43′38.68″N, 87°41′56.81″W, coralline rock, 2 m, Mar 4, 1998, colls. S. Salazar-Vallejo, L. Carrera-Parra. ECOSUR-S73, 6 specimens, reef lagoon near to back reef, 18°42′34.01″N, 87°42′31.22″W, coralline rock, 1.5 m, Jan 9, 2001, colls. P. Salazar-Silva, J. Bastida-Zavala, M. Tovar-Hernández, S. Salazar-Vallejo, L. Carrera-Parra. ECOSUR-S74, 1 specimen, fore reef, 18°42′43.32″N, 87°42′22.51″W, coralline rock, 15 m, Jun 6, 1998, coll. M. Ruiz-Zárate. ECOSUR-S75, 2 specimens, reef lagoon, 18°42′36.23″N, 87°42′31.60″W, coralline rock, 1.5 m, Dec 1, 2000, colls. S. Salazar-Vallejo, L. Carrera-Parra. ECOSUR-S76, 2 specimens, reef crest, 18°43′06.17″N, 87°42′14.17″W, coralline rock, 1 m, Jul 21, 1998, colls. S. Salazar-Vallejo, L. Carrera-Parra. ECOSUR-S77, 5 specimens, back reef, 18°42′31.30″N, 87°42′30.39″W, coralline rock, 2 m, Mar 22, 2000, colls. S. Salazar-Vallejo, L. Carrera-Parra. ECOSUR-S78, 1 specimen, reef lagoon, 18°42′36.17″N, 87°42′32.65″W, coralline rock, 1.5 m, Mar 21, 2000, colls. J. Bastida-Zavala, P. Salazar-Silva. ECOSUR-S79, 5 specimens, old wooden pier, 18°42′41.95″N, 87°42′35.98″W, fouling, 1 m, Feb 24, 2001, colls. P. Salazar-Silva, J. Bastida-Zavala, M. Tovar-Hernández, S. Salazar-Vallejo, L. Carrera-Parra, L. Harris. ECOSUR-S80, 8 specimens, old wooden pier, 18°42′41.95″N, 87°42′35.98″W, fouling, 1 m, Mar 18, 2001, colls. P. Salazar-Silva, J. Bastida-Zavala, M. Tovar-Hernández, S. Salazar-Vallejo, L. Carrera-Parra. ECOSUR-S81, 14 specimens, reef lagoon, 18°43′22.73″N, 87°42′03.08″W, coralline rock, 1.5 m, Mar 28, 2001, colls. L. Carrera-Parra, M. Londoño-Mesa, S. Salazar-Vallejo. ECOSUR-S82, 1 specimen, reef lagoon, 18°43′25.14″N, 87°42′01.75″W, coralline rock, 1.5, Jan 10, 2001, colls. L. Carrera-Parra, M. Londoño-Mesa, S. Salazar-Vallejo. ECOSUR-S83, 12 specimens, reef lagoon, 18°43′24.93″N, 87°42′02.95″W, coralline rock, 1 m, Jan 19, 2001, colls. P. Salazar-Silva, J. Bastida-Zavala, M. Tovar-Hernández, S. Salazar-Vallejo, L. Carrera-Parra. ECOSUR-S84, 2 specimens, reef lagoon, 18°43′24.93″N, 87°42′02.95″W, coralline rock, 1 m, Nov 30, 2000, colls. S. Salazar-Vallejo, L. Carrera-Parra. ECOSUR-S85, 13 specimens, reef lagoon, 18°43′21.01″N, 87°42′04.28″W, coralline rock, 1.5 m, Feb 24, 2001, colls. P. Salazar-Silva, J. Bastida-Zavala, M. Tovar-Hernández, S. Salazar-Vallejo, L. Carrera-Parra, L. Harris. ECOSUR-S93, 1 specimen, punta Rio Bermejo, 18°41′07.53″N, 87°43′05.83″W, corralline rock, 1 m, May 17 2002, colls. S. Salazar-Vallejo, M. García-Madrigal. ECOSUR-S94, 16 specimens, punta Rio Indio, 18°48′29.63″N, 87°39′58.82″W, coralline rock, 1.7 m, Mar 17, 2001, coll. L. Carrera-Parra. Xahuayxol, ECOSUR-S58, 8 specimens, reef lagoon, 18°30′11″N, 87°45′29″W, coralline rock, 2 m, Jun 1 1997, colls. S. Salazar-Vallejo, L. Carrera-Parra. ECOSUR-S59, 1 specimen, back reef, 18°30′11.45″N, 87°45′21.46″W, coralline rock, 12 m, Oct 30 1997, colls. R. Saenz-Morales, S. Salazar-Vallejo, L. Carrera-Parra. ECOSUR-S60, 4 specimens, reef lagoon, 120 m off coast, 18°30′41.34″N, 87°45′24.63″W, coralline rock, 1.5 m, Oct 31 1997, colls. S. Salazar-Vallejo, L. Carrera-Parra. ECOSUR-S61, 11 specimens, reef lagoon, 18°30′39.77″N, 87°45′24.80″W, coralline rock, 1.8 m, Jun 4, 1998, colls. S. Salazar-Vallejo, L. Carrera-Parra. ECOSUR-S68, 3 specimens, reef lagoon, 18°30′39.04″N, 87°45′25.09″W, coralline rock, 1.7 m, Sep 27, 1996, colls. S. Salazar-Vallejo, L. Carrera-Parra. ECOSUR-S95, 21 specimens, reef lagoon, 18°30′12.46″N, 87°45′29.79″W, coralline rock, 2 m, Jun 1, 1997, colls. S. Salazar-Vallejo, L. Carrera-Parra. ECOSUR-S97, 1 specimen, 100 m off coast, 18°30′41.43″N, 87°45′25.16″W, coralline rock, 2 m, Sep 28, 1996, coll. S. Salazar-Vallejo, L. Carrera-Parra. ECOSUR-S98, 1 specimen, reef lagoon, 18°30′12.46″N, 87°45′29.79″W, coralline rock, 2 m, Jun 1, 1997, colls. S. Salazar-Vallejo, L. Carrera-Parra. ECOSUR-S99, 6 specimens, reef lagoon, 18°30′13.71″N, 87°45′31.50″W, coralline rock, 1 m, Jun 2 1998, colls. S. Salazar-Vallejo, L. Carrera-Parra. ECOSUR-S100, 2 specimens, 100 m off coast, 18°30′15.08″N, 87°45′30.98″W, in sediment with Thalassia testudinum, 2 m, Sep 27, 1996, colls. S. Salazar-Vallejo, L. Carrera-Parra. Xcalak. ECOSUR-S101, 9 specimens, back reef, 18°15′50.40″N, 87°49′31.12″W, coralline rock, 1.7 m, Oct 25, 2002, colls. S. Salazar-Vallejo, L. Carrera-Parra, P. Salazar-Silva, M. Londoño-Mesa. Nicaragua, Nicaraguan Caribbean. UMML 000, 1 specimen, R/V Pillsbury, Cruise 7101, sta. 1338, 12°52′00″N, 82°35′17.98″W, hillocks or low rounded mounds, 28 m, Jan 29, 1971, coll. G. Voss. Colombia, Providence Island. UMML 000, 1 specimen, R/V Pillsbury, Cruise 7101, sta. 1349, 13°33′00″N, 81°28′00″W, patch reef slightly southwest of Low Island surrounded by sand bottom, 3 m, Jan 30, 1971. Dominican Republic. UMML 000, 2 specimens. R/V Pillsbury, Cruise 7006, sta. 1272, off Cabo Rojo, 17°52′41.98″N, 71°41’12.01″W, 20-27 m, Jul 17, 1970, coll. J. Staiger. Turks and Caicos. UMML 000. 1 specimen, Pillsbury Cruise 7106, sta. 1423, 21°40′59.98″N, 71°22′59.98″W, Jul 19, 1971. Saint Martin, Îlet de L´embouchure. UF 329, 1 specimen, 18°04′1.2″N, 63°00′39.6″W, reef flat lagoon with seagrass, 1 m, Apr 9, 2012, colls. G. Paulay, J. Slapcinsky, M. Bemis. UF 330, 5 specimens, 18°04′01.20″N, 63°00′43.20″W, reef flat lagoon with seagrass, 0-1 m, Apr 9, 2012, colls. G. Paulay, J. Slapcinsky, M. Bemis. UF 338. 1 specimen, 18°04′01.20″N, 63°00′43.20″W, reef, 1 m, Apr 17, 2012, coll. A. Anker. UF 340, 1 specimen, 18°04′01.20″N, 63°00′43.20″W, reef, 1 m, Apr 17, 2012, coll. A. Anker. UF 347, 1 specimen, 18°04′01.20″N, 63°00′43.20″W, reef, 0-1 m, Apr 17, 2012, colls. G. Paulay, J. Slapcinsky, A. Anker. UF 359, 3 specimens, steep rubbly reef slope, 2-10 m, Apr 22, 2012, colls. G. Paulay, F. Michonneau. Mont Vernon, UF 332, 1 specimen, Little Key, 18°06′00″N, 63°01′19.20″W, seagrass, sand, rocks, 0-2 m, Apr 11, 2012, coll. J. Slapcinsky. Grande Caye, UF 353, 1 specimen, 18°06′43.20″ N, 63°01′08.40″W, coral rocks, 3 m, Apr 20, 2012, coll. A. Anker. Rocher Créole. UF 352. 1 specimen, 18°07′04.80″N, 63°03′21.60″W, reef, 9 m, Apr 18, 2012, colls. G. Paulay, J. Slapcinsky, M. Bemis. N. side of St. Martin, UF 336, 1 specimen, 18°07′48″N, 63°00′18″W, canyon with sponges, in algae, 13 m, Apr 11, 2012, coll. R. Renoux. Île Tintammare, UF 361, 3 specimens, Chicot, windward side of the island, 18°06′7.2″N, 62°58′58.80″W, reef, in rubble, 12-15 m, Apr 23, 2012, coll. G. Paulay. Caye Verte, UF 364, 1 specimen, 18°05′24″N, 63°00′43.2″W, reef with sand and grass, 7 m, Apr 25, 2012, coll. M. Bemis. Panama. Bocas del Toro. UF 126, 1 specimen, coll. C. Meyer. UF 485. 1 specimen, Punta Puebla, 9°22′01.20″N, 82°17′27.60″W, May 16, 2016, colls. M. Leray, F. Michonneau, R. Lasley. UF 486, 1 specimen, Punta Juan, 9°18′03.60″N, 82°17′38.40″W, May 16, 2016, coll. R. Lasley. UF 494, 1 specimen, Runway, 9°20′31.20″N, 82°15′36″W, May 23, 2016, colls. M. Leray, F. Michonneau, R. Lasley. UF 499. 1 specimen, Marina, 9°19′51.60″N, 82°14′38.40″W, May 28, 2016, colls. M. Leray, F. Michonneau, R. Lasley. UF 500, 2 specimens, Marina, 9°19′51.60″N, 82°14′38.40″W, May 28, 2016, colls. M. Leray, F. Michonneau, R. Lasley. UF 501, 1 specimen, Cayo Hermanas, 9°16′04.80″N, 82°21′07.20″W, May 30, 2016, colls. M. Leray, F. Michonneau, R. Lasley. UF 503, 1 specimen. Cayo Hermanas, 9°16′04.80″N, 82°21′07.20″W, May 31, 2016, colls. M. Leray, F. Michonneau, R. Lasley. UF 541, 1 specimen, Salt Creek, 9°16′48″N, 82°06′07.20″W, outer reef, Agaricia reef, 4-4.5 m, May 22, 2016, colls. M. Leray, F. Michonneau, R. Lasley. UF 542, 1 specimen, Runway, 9°20′31.20″N, 82°15′36″W, May 23, 2016, colls. M. Leray, F. Michonneau, R. Lasley, 4–4.5 m, lagoon fringing reef, Agaricia reef.

Redescription

Male specimen from St. Martin, West Indies (UF 332).

External anatomy. Trunk 14 mm in length (Fig. 1A); light brown with some darker patches; papillae dark brown and light brown distributed randomly, most of them on the dorsal region, scarce ventrally; papillae conglomerated in anal region (Fig. 1D) and caudal region (Fig. 1F), most dispersed in median region (Fig. 1E).

Figure 1 Papillar morphology on Phascolosoma (Phascolosoma) varians Keferstein, 1865.

Regional variation in the patterns of papillae along the body are shown for specimen UF 332, from Saint Martin, West Indies. (A) Adult body plan in dorsal view with anterior to the top. (B) Papillae from introvert region with light transversal bands. (C) Papillae from introvert. (D) Papillae from anal region of trunk. (E) Papillae from median region of trunk. (F) Papillae from caudal region of trunk. Scale bars = (A), 2 mm; (B and C), 0.3 mm; (D), 1 mm; (E and F), 0.5 mm. White arrow indicates the location of the anus.

Introvert 10 mm length (Fig. 1A), dark brown, uniform color ventrally and dorsally, with ill-defined transversal bands light brown color (Fig. 1B), proximal introvert with similar color pattern as trunk, extensible collar small. Introvert papillae dark brown, smaller than those of trunk (Fig. 1C), located between rings of hooks, starting from the first ring (Fig. 2H), these are likely secretory or sensory papillae. Nuchal organ with wavy contour.

Figure 2 Anatomical characters of Phascolosoma (Phascolosoma) varians Keferstein, 1865.

Micrographs of hook, muscle and papillar anatomy are shown for specimen UF 332, from Saint Martin, West Indies, unless stated. (A) Distal hooks, (B) hooks of median region of introvert and (C) proximal hooks. (D) Internal anatomy of a dissected specimen, with anterior to the top. (E) Posterior attachment site of a ventral retractor muscles (VRM) associated with longitudinal muscle bands 1–9; specimen ECOSUR-S94 from the Mexican Caribbean. (F) SEM image of hook from anterior region of the introvert; specimen UF359 from St. Martin, West Indies. (G) SEM image of hooks from median region of the introvert. (H) SEM image showing stages of hook growth; from left right: first seven rings of introvert hooks, arrows show individual papillae between rows of hooks. (I) SEM image of papillae from proximal introvert (J) SEM image of papilla from median region of the introvert; specimen UF 26.5 from Florida. (K) SEM image of papilla from distal introvert. Abbreviations: DRM, dorsal retractor muscles; IN, intestine; LMB, longitudinal muscle bands; N, nephridia; SM, spindle muscle; SEM, scanning electron microscope; VRM, ventral retractor muscles. Scale bars = (A–C), 30 μm; (F, G, I and J), 20 μm; (H), 50 μm; (K), 5 μm. Photographer: Luis F. Carrera-Parra.

Hooks laterally compressed, arranged in 60 complete rings and some incomplete rings, probably due to abrasion. Rings followed by a zone with scattered hooks. Distal hooks (Figs. 2A and 2F) with an angle of 90° between line X and Y; length of distal tip never projecting beyond the base of the hook; secondary tooth rounded; internal clear streak (apical canal) expanded near to midpoint of vertical and middle horizontal portions of hook. Hooks of median region with a larger secondary tooth (Figs. 2B and 2G), proximal hooks (Fig. 2C) with principal tooth smaller than its base, almost 25% less. Distal hooks with external border bent squarely; hooks of the median region of the introvert with progressively rounder bent border; proximal hooks with evenly rounded external border.

SEM revealed growth stages of hooks and papillae (Fig. 2H); smaller in the distal introvert and larger at the proximal introvert. Introvert papillae with three stages of development. First stage: the smallest, spherical with a ring of short apical protrusions (Fig. 2K) “dome shape” (fide Cutler, 1994). Second stage: medium size, appearance of two units, the smallest with a ring of short apical protrusions, and a broad base (Fig. 2J) “mammillate form” (fide Cutler, 1994). Third stage: largest, conical (Fig. 2I) “cone shape” (fide Cutler, 1994).

Internal anatomy (Fig. 2D). A pair of nephridia occupying 80% of trunk length, open at the same level as anus. Longitudinal musculature divided into 23 individual and anastomosed bands in the median trunk. Two pairs of retractor muscles; ventral pair attached to 8 longitudinal bands starting from the third band after ventral nerve cord (Fig. 2E), dorsal pair attached to 5 longitudinal bands starting from the fifth band after ventral nerve cord. Contractile vessel without swelling or villi. Spindle muscle attached posteriorly.

Habitat. In coralline rock and hard bottom, 1–30 m depth.

Distribution. Greater Caribbean from Florida to Venezuela.

Remarks. Keferstein (1865) described Phascolosoma (Phascolosoma) varians from St. Thomas, West Indies and P. (P.) nigrescens from Fiji. Although Keferstein’s descriptions were well illustrated, Selenka (1883) produced an improved set of drawings from the type materials (Fig. 3). Keferstein (1865) recognized each species as follows: Phascolosoma varians with a body three to four times as long as thick; introvert as long or longer than the body; closely spaced rows of hooks, highly variable in number (12–90), which often only cover the anteriormost part of the trunk; hooks very broad, with an upper right-angled tip (0.072 mm high, 0.092 mm wide); with 20–28 short tentacles, standing in two rows at the side; longitudinal muscles about 30, but in many cases anastomosed as longitudinal strands; contractile vessel simple, without lateral sags; nephridia very long, attached in the anterior third by a mesentery. Phascolosoma nigrescens has a trunk about four times as long as thick; introvert longer than trunk; numerous hooks forming rings situated very close to each other in the anterior end of the trunk; hooks flattened with an lower right-angled tip (0.084 mm high, 0.084 mm wide); over 20 tentacles in several rows; muscles separated in about 24 longitudinal strands with few anastomosed bands; contractile vessel on the esophagus with many small, lateral sags; nephridia attached along almost their entire length by a wide mesentery.

Figure 3 Hooks and papillae of Phascolosoma varians and Phascolosoma nigrescens.

From drawings of (A–D) Phymosoma varians, Barbados and (E–H) Phymosoma nigrescens, Fiji Islands, by Selenka (1883). (A) Trunk hook. (B) Papilla between the hook rings on introvert. (C) Papilla from side in longitudinal section. (D) Skin of the middle of the body with domed papillae. (E) Trunk hook. (F) Middle of papilla in “G” from above. (G) Papilla between the scattered introvert hooks. (H) Domed body papilla at 140× magnification. Both species originally described by Keferstein (1865).

Cutler & Cutler (1990) reviewed the subgenus Phascolosoma (Phascolosoma), and designated P. (P.) varians, together with nine other species from different regions of the world, as junior synonyms of P. (P.) nigrescens. At that time, Cutler & Cutler (1990) suggested there were “no consistent differences” between these two species. Their decision appears to be primarily based upon variation associated with hook morphology. According to Cutler & Cutler (1990): “One possible hypothesis is that hook morphology is determined by more than one pair of genes and that allelic frequencies vary from place to place. The alleles for sharp angle and large secondary tooth occur at a high frequency in the Caribbean and a low frequency in the Indo-West Pacific… P. varians is the junior name because it was described later on the page”. No molecular evidence was ever provided in support of the allelic hypothesis, or any other genetic differences supporting proposed junior synonyms. Subsequently, Cutler (1994) published his synonymy upon further consideration of the morphological differences between P. (P.) nigrescens and P. (P.) varians.

Herein, reexamination and redescription of P. (P.) varians, revealed clear morphological differences concerning its previously designated senior synonym. The most important features that distinguish these two species include the shape of the hooks, the attachment of nephridia to the body wall, and morphology the contractile vessel. Phascolosoma (P.) varians has hooks with a rounded secondary tooth; the base of the hook is broader than high; most of the anterior hooks (Figs. 2A and 2F) possess a distal tip at a 90° angle with respect to the perpendicular axial line of the hook; the contractile vessel is simple; nephridia are attached to body wall in the anterior third. P (P.) nigrescens has hooks with a square secondary tooth; the base of hook is as broad as high; most of the anterior hooks possess a distal tip with less than a 90° angle with respect to the perpendicular axial line of the hook; a contractile vessel on the esophagus with many small, lateral sags; nephridia are attached almost along their entire length.

The wavy contour of the nuchal organ and specific attachments of the retractor muscles were not described for both species by Keferstein in 1865, nevertheless, these features are now described. Keferstein (1865) refers to “tentacles in two rows or many rows”, which may explain the wavy contour of the nuchal organs I observed. The differences between the number of tentacles is not useful for separating these species because they appear to vary with the development stage of the specimen, and that variation overlaps between these species. Because of the difficulty of establishing the exact number of longitudinal muscle bands, this characteristic should be considered cautiously. Additionally, the number of rings of hooks is variable between these species, and the loss of rings is not uncommon. Papillae are also inappropriate characters for distinguishing these species as their differences are minor across most of the body surface.

Phascolosoma (Phascolosoma) granulatum was included for comparison in the molecular analyses. The species can be easily distinguished from P. (P.) varians for having hooks with a basal triangle and lacking bands of pigmentation in the introvert, while P. (P.) varians lacks a basal triangle in the hooks and the bands of pigmentation are conspicuous.

Molecular analyses

Herein, the synonymy of Phascolosoma (P.) varians with Phascolosoma (P.) nigrescens is rejected due to morphological differences. Additionally, there is a high genetic divergence between nucleotide sequences within their respective cytochrome c oxidase subunit 1 (COI) genes, which supports the morphological data.

The first three sequences (Fig. 4) (top-down) correspond to Phascolosoma (P.) varians from the Greater Caribbean. This species had been determined as Phascolosoma (P.) nigrescens by past authors and registered as such it in GenBank. These sequences were grouped with a low intraspecific variation of 2.6%. The localities of those sequences are Florida and Barbados; preserved specimens from the same localities were revised morphologically to support the correct identification of the species.

Figure 4 Maximum likelihood analysis of cytochrome c oxidase subunit 1 (COI) DNA.

Phascolosoma (P.) varians from the Greater Caribbean shows a clear genetic distinction from all specimens identified as Phascolosoma (P.) nigrescens from eastern and western regions of the Pacific Ocean. Tree reconstruction was generated from a ClustalW alignment of COI sequences amplified from 11 sipunculan specimens. Species names (specimens) and GenBank accession numbers are listed at the branch tips. Individuals and conspecific groups are marked with vertical bars, and their corresponding geographic regions, to the right of species names. This analysis was performed using Tamura 3-parameter with a discrete Gamma distribution with five rate categories, assuming a certain fraction of sites is evolutionarily invariable (T92 + G + I).

The two sequences of Phascolosoma (P.) nigrescens from New Caledonia constitute a group clearly separated from Phascolosoma (P.) varians from the Greater Caribbean (genetic mean distance 24%). There are two crucial facts to consider. First: New Caledonia is the closest locality to Fiji, the type locality of P. (P.) nigrescens. The high genetic divergences between the specimens from New Caledonia and those from the Greater Caribbean supports the morphological differences between both species and reinforce the reinstatement of P. (P.) varians. Second: the intraspecific variation between both sequences is very high (18%), almost the same value of the interspecific variation of the other groups in this analysis. This value suggests that it is highly likely that these two sequences represent different morphotypes, and either one of them would correspond with Phascolosoma (P.) nigrescens.

The values of the genetic distance between Phascolosoma (P.) nigrescens from Israel, South Africa and, Broome, Australia, varies from 18% to 26% regarding Phascolosoma (Phascolosoma) varians from the Greater Caribbean. These values are similar to the results of Silva-Morales et al. (2019), where they found a genetic distance of 19% between Antillesoma antillarum (Greater Caribbean) and A. mexicanum (Southern Mexican Pacific). The present analysis reveals that Phascolosoma (Phascolosoma) nigrescens is a species complex resulting from the incipient morphological analysis. This study shows that Phascolosoma (P.) varians from the Greater Caribbean is well differentiating morphologically and genetically of Phascolosoma (P.) nigrescens; however, a detailed morphological revision of this species complex is needed.

Discussion

The genetic analysis suggests the following considerations: (1) the specimens from Israel, South Africa and Broome, Australia identified as Phascolosoma (P.) nigrescens correspond to different species. It is very likely that other species from those regions, considered synonyms, would need to be reinstated following redescription, or described and established as new species. (2) It will be necessary to resolve a potential species complex of Phascolosoma (P.) nigrescens by combining molecular and morphological data. Neither one of these considerations were resolved in the present study, as they were beyond the original scope of this investigation.

The assumed wide distribution of Sipuncula taxa is attributed to the development of the teleplanic larvae in some species, such as Phascolosoma (Phascolosoma) varians. Although it is well accepted that the free-swimming larval stage with its prominent locomotive organ (known as a metatroch) confers the ability to disperse, allowing an increase in geographic range and providing for genetic exchange between populations (Rice, 1981), it is not necessarily the rule. Staton & Rice (1999) described the case of a species with teleplanic larvae with a limited distribution. They found distinct genetic differences within and between larval and adult stages of Apionsoma (A.) misakianum (Ikeda, 1904) from northern and southern regions of Florida and the Bahamas using allozymes. They did not find any indication of “hybrids” occurring between them, suggesting a potential oceanic boundary was present between populations with teleplanic pelagosphera larvae, and thus a possible Apionsoma species-complex in the region. They did not perform a morphological analysis of the adults at that time, but the larve exhibited two distinct coloration patterns.

Kawauchi & Giribet (2010) rejected the cosmopolitanism of Phascolosoma (P.) perlucens Baird, 1868, by analyzing molecular and morphological data of specimens from many localities around the world. They detected four different lineages, and identified variation in hook morphologies between different localities that correlated with a high genetic diversity between populations. Also, their results suggested a probable lack of gene flow between the geographically distinct lineages. Schulze et al. (2012) analyzed molecular sequence data and developmental features of three “cosmopolitan” species, Phascolosoma (P.) agassizii Keferstein, 1866, Thysanocardia nigra (Ikeda, 1904), and Themiste (T.) pyroides (Chamberlin, 1920). For each one of the three species, they found significant differences between previously assumed con-specific populations from the Sea of Japan and the Northeast Pacific region, with respect to egg size, developmental mode and developmental timing. The populations of all three species were remarkably distinct genetically and suggested that gene flow between the two regions was extremely unlikely. Furthermore, Kawauchi & Giribet (2014) analyzed the genetic data from four genes of Sipunculus (S.) nudus Linnaeus, 1766 with specimens from multiple localities worldwide. As with Phascolosoma perlucens (see above), these two investigators again found high levels of genetic differentiation between distantly related populations, suggesting in this case there were five distinct lineages, three of which could be distinguished morphologically. In the last two studies, neither a new species was described, nor an available name reinstated.

Conclusions

High genetic divergence between specimens identified as P. P. nigrescens from the Greater Caribbean and a region close to its type locality correlate with morphological differences found between P. (P.) varians and P. (P.) nigrescens. Herein, the synonymy of Phascolosoma (P.) varians with Phascolosoma (P.) nigrescens is rejected, and as a consequence, Phascolosoma (P.) varians is reinstated.

Based upon these findings, and other studies, some of which are discussed here, the diversity of sipunculans is most likely underestimated. Thus, a combination of morphological and molecular data, along with other important information from the fields of development, ecology and oceanography, will help us determine a more realistic number of extant sipunculans species worldwide.

I would like to thank Luis F. Carrera-Parra (ECOSUR) who contributed greatly with his recommendations and comments to improve the manuscript, and his production of SEM photos. Thanks to Nancy Voss (UMML) and Gustav Paulay (UF) who kindly lent the specimens used in this study. Thanks to Sergio I. Salazar-Vallejo (ECOSUR) and Luis F. Carrera-Parra who provided specimens from the Collection of Benthos of El Colegio de la Frontera Sur, Chetumal, Quintana Roo, Mexico. Thanks to Gerardo Flores-Taboada for his comments and for reviewing the translation. Thanks to Mario H. Londoño-Mesa and to an anonymous reviewer for their suggestions to improve this manuscript.

Additional Information and Declarations

Competing Interests

Author Contributions

Data Availability

The authors declare that they have no competing interests.

Itzahi Silva-Morales conceived and designed the experiments, performed the experiments, analyzed the data, prepared figures and/or tables, authored or reviewed drafts of the paper, and approved the final draft.

The following information was supplied regarding data availability:

The reviewed specimens of Phascolosoma varians are available in the following scientific collections:

(1) Marine Invertebrate Museum (UMML), Rosenstiel School of Marine and Atmospheric Science, University of Miami;

(2) Invertebrate Collections of the Florida Museum of Natural History (UF), University of Florida;

(3) Reference Collection of Benthos (ECOSUR) of El Colegio de la Frontera Sur, Chetumal, Mexico.

The catalog numbers are listed in the “Material examined”.

The COI sequences are available at GenBank: DQ300139, DQ300142, AY161122, DQ300143, JN865121, JN865122, DQ300140, DQ300141, DQ300138, JQ904338, JQ904337.

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
