# Peer review of "Reinstatement of Phascolosoma (Phascolosoma) varians Keferstein, 1865 (Sipuncula: Phascolosomatidae) based on morphological and molecular data"

_PeerJ, doi:10.7717/peerj.10238_

## Round 0.1 · original submission · Major Revisions

Dear author,

Many thanks for your submission to Peer J. Although I have decided for Major Revision, most changes should be fairly easy to resolve. Both reviewers have made an incredible job revising the language and content of your manuscript.

Looking forward to receiving your revised manuscript.

Reviewer 1 ·

Basic reporting

Reinstatement of Phascolosoma (Phascolosoma) varians Keferstein, 1865 (Sipuncula: Phascolosomatidae) based on morphological and molecular data

Itzahí Silva-Morales


Decision: Review of Article ID 51760 is highly favorable, and the Reviewer recommends this manuscript for publication in PeerJ. This manuscript is important to the specific field of sipunculan taxonomy, and a welcome contribution to the field of taxonomy in general.


1. Basic Reporting:

There are issues with use of the English language throughout the manuscript, which includes elements of grammar, word usage, word agreement, sentence structure, insertion of terms, and the clarity of particular statements. These are expected from an Author with English as a second language. Therefore, because of the language issues, the Reviewer has taken a focused interest in improving the manuscript in a non-conventional manner. There are many instances where a line-by-line review of grammatical errors and statement clarity issues would require considerable time, and many pages of review notes to be submitted. Therefore this Reviewer has edited the Word.docx version of the original manuscript with Tracked Changes throughout most of the text, and has placed a number of comments and suggestions along the margin.

Note: the Author may, or may not, accept or use the recommended Tracked Changes, but it is the Reviewer’s opinion that such recommendations will significantly improve the manuscript.

Literature sources are sufficient for this study and the extensive background required to make the case for reinstatement of a species that was previously synonymized with another species. Author should recheck the References section for accuracy and formatting, and see comments in the margin of the Tracked Changes document (e.g. Quatrefages vs. de Quatrefages; Broeke vs. ten Broeke, etc.). Regarding the Material Examined section, Reviewer did not review this.

The structure of the manuscript meets professional standards and general PeerJ formatting. However, the figure captions have also been extensively edited with Tracked Changes by the Reviewer to comply with PeerJ figure caption formats, and they were edited for a more specific, detailed and thorough description of the primary data presented in each of the four figures.

Experimental design

2. Experimental design:

This manuscript is original research, and it meets the Aims & Scope of PeerJ, particularly in the areas of Biological Sciences, as a data-driven Research Article, and on its methodological soundness. This manuscript goes far in its effort to examine previous research, provide a comprehensive set of specimen observations, with corresponding genetic support for those observations, and aims to correct notable errors by previous workers on this particular topic.

The question is clear, the methods utilized to answer the question are appropriate, and the work therein builds upon previous efforts to answer similar concerns by other investigators who have pursued this same question. There certainly is a gap in our knowledge of species distinctions and the connectivity of assumed con-specific populations. More of this work will be required going forward. And, the Author has made a strong case for such efforts here.

Note: The Reviewer would like to see the Author develop a more comprehensive Introduction section by adding pertinent background on the relevance of using COI sequences (e.g. a Barcode gene) for identifying and distinguishing species within the target group, and other metazoan groups outside Sipuncula, with examples. And, address whether COI alone is sufficient, or should it be complemented with other genetic markers to strengthen the results. Additionally, there should be more introductory information on sipunculan life histories, in particular the types of larvae within the clade, and the limitations of different larval modes (e.g. lecithotrophic, planktotrophic, teleplanic) to disperse and connect species populations, as suggested by Cutler & Cutler (1990), who have made this manuscript necessary. These two issues, larvae as vectors of connectivity, and genetic markers of species identification, should then be revisited in the Discussion section with more detail than first provided by the Author.

The Reviewer finds no outstanding issues with technical standards as presented, and the methods described are both appropriate and replicable, as required for similar studies on sipunculan taxonomy that must be performed in the future. As noted above, the Reviewer has performed edits, added comments and posed questions that will clarify particular methods and data presentation materials – to improve access to specific details by the Author’s readership.

Validity of the findings

3. Validity of the Findings:

There are no obvious concerns regarding acceptability of the results, completeness of the data provided or the conclusions made. Although, the Reviewer thinks that there are more conclusions from this investigation than those which are stated in the Conclusions section.

The Reviewer suggests that the Author add support values (X.X%) to the COI tree (Figure 4), and provide more discussion as to how and why those values are acceptable measures, confirming the proposed status of Phascolosoma (Phascolosoma) varians as “reinstated”. For instance, what does the literature state as ‘acceptable’ for separate species? This is important because no molecular data was used to support the original synonymies by Cutler and Cutler (1990).

Note: In the Remarks section, Keferstein’s descriptions are interesting and relevant, but they are only general. The Author should highlight the differences between species and then use a separate paragraph to ‘remark’ on the specific differences as stated by Stephen & Edmonds (1972), to perhaps complement the Systematics section (lines 134–157) within the Results.

The Author should also mention the condition and dimensions of the fixed material if known.

Final note: use of the term ‘reinstated’ does not need to be written with every listing of the species, as the purpose of this manuscript is very clear. Please see the edits to figure captions by the Reviewer. The terms, ‘reestablished’ and ‘restored’ are not helpful and were removed.

Final comment/question to Author: will this investigation have any anticipated impact on the current relocation of Sipuncula to the basal branches within the annelid radiation? At some time in the future, sipunculan taxonomic revisions, including genetic comparisons with several polychaete annelid sister taxa, will require similar efforts to distinguish sipunculan species. Does your work anticipate any role in this larger effort? If so, how? If not, why not?

And, although this manuscript focuses on a species-specific topic of clarification, does the Author have any future directions for the next species targets within Sipuncula?

Additional comments

Reinstatement of Phascolosoma (Phascolosoma) varians Keferstein, 1865 (Sipuncula: Phascolosomatidae) based on morphological and molecular data

Itzahí Silva-Morales


Decision: Review of Article ID 51760 is highly favorable, and the Reviewer recommends this manuscript for publication in PeerJ. This manuscript is important to the specific field of sipunculan taxonomy, and a welcome contribution to the field of taxonomy in general.


1. Basic Reporting:

There are issues with use of the English language throughout the manuscript, which includes elements of grammar, word usage, word agreement, sentence structure, insertion of terms, and the clarity of particular statements. These are expected from an Author with English as a second language. Therefore, because of the language issues, the Reviewer has taken a focused interest in improving the manuscript in a non-conventional manner. There are many instances where a line-by-line review of grammatical errors and statement clarity issues would require considerable time, and many pages of review notes to be submitted. Therefore this Reviewer has edited the Word.docx version of the original manuscript with Tracked Changes throughout most of the text, and has placed a number of comments and suggestions along the margin.

Note: the Author may, or may not, accept or use the recommended Tracked Changes, but it is the Reviewer’s opinion that such recommendations will significantly improve the manuscript.

Literature sources are sufficient for this study and the extensive background required to make the case for reinstatement of a species that was previously synonymized with another species. Author should recheck the References section for accuracy and formatting, and see comments in the margin of the Tracked Changes document (e.g. Quatrefages vs. de Quatrefages; Broeke vs. ten Broeke, etc.). Regarding the Material Examined section, Reviewer did not review this.

The structure of the manuscript meets professional standards and general PeerJ formatting. However, the figure captions have also been extensively edited with Tracked Changes by the Reviewer to comply with PeerJ figure caption formats, and they were edited for a more specific, detailed and thorough description of the primary data presented in each of the four figures.


2. Experimental design:

This manuscript is original research, and it meets the Aims & Scope of PeerJ, particularly in the areas of Biological Sciences, as a data-driven Research Article, and on its methodological soundness. This manuscript goes far in its effort to examine previous research, provide a comprehensive set of specimen observations, with corresponding genetic support for those observations, and aims to correct notable errors by previous workers on this particular topic.

The question is clear, the methods utilized to answer the question are appropriate, and the work therein builds upon previous efforts to answer similar concerns by other investigators who have pursued this same question. There certainly is a gap in our knowledge of species distinctions and the connectivity of assumed con-specific populations. More of this work will be required going forward. And, the Author has made a strong case for such efforts here.

Note: The Reviewer would like to see the Author develop a more comprehensive Introduction section by adding pertinent background on the relevance of using COI sequences (e.g. a Barcode gene) for identifying and distinguishing species within the target group, and other metazoan groups outside Sipuncula, with examples. And, address whether COI alone is sufficient, or should it be complemented with other genetic markers to strengthen the results. Additionally, there should be more introductory information on sipunculan life histories, in particular the types of larvae within the clade, and the limitations of different larval modes (e.g. lecithotrophic, planktotrophic, teleplanic) to disperse and connect species populations, as suggested by Cutler & Cutler (1990), who have made this manuscript necessary. These two issues, larvae as vectors of connectivity, and genetic markers of species identification, should then be revisited in the Discussion section with more detail than first provided by the Author.

The Reviewer finds no outstanding issues with technical standards as presented, and the methods described are both appropriate and replicable, as required for similar studies on sipunculan taxonomy that must be performed in the future. As noted above, the Reviewer has performed edits, added comments and posed questions that will clarify particular methods and data presentation materials – to improve access to specific details by the Author’s readership.


3. Validity of the Findings:

There are no obvious concerns regarding acceptability of the results, completeness of the data provided or the conclusions made. Although, the Reviewer thinks that there are more conclusions from this investigation than those which are stated in the Conclusions section.

The Reviewer suggests that the Author add support values (X.X%) to the COI tree (Figure 4), and provide more discussion as to how and why those values are acceptable measures, confirming the proposed status of Phascolosoma (Phascolosoma) varians as “reinstated”. For instance, what does the literature state as ‘acceptable’ for separate species? This is important because no molecular data was used to support the original synonymies by Cutler and Cutler (1990).

Note: In the Remarks section, Keferstein’s descriptions are interesting and relevant, but they are only general. The Author should highlight the differences between species and then use a separate paragraph to ‘remark’ on the specific differences as stated by Stephen & Edmonds (1972), to perhaps complement the Systematics section (lines 134–157) within the Results.

The Author should also mention the condition and dimensions of the fixed material if known.

Final note: use of the term ‘reinstated’ does not need to be written with every listing of the species, as the purpose of this manuscript is very clear. Please see the edits to figure captions by the Reviewer. The terms, ‘reestablished’ and ‘restored’ are not helpful and were removed.

Final comment/question to Author: will this investigation have any anticipated impact on the current relocation of Sipuncula to the basal branches within the annelid radiation? At some time in the future, sipunculan taxonomic revisions, including genetic comparisons with several polychaete annelid sister taxa, will require similar efforts to distinguish sipunculan species. Does your work anticipate any role in this larger effort? If so, how? If not, why not?

And, although this manuscript focuses on a species-specific topic of clarification, does the Author have any future directions for the next species targets within Sipuncula?


4. General Comments:

This manuscript is likely one of many more to come from the collective efforts of investigators to reexamine and to correct the extensive synonymization of sipunculan taxa, as most recently published by Cutler in 1994. Cutler changed the number of recognized species of sipunculans by ~ 171 species, from an already low-species-number taxon. The Author has referenced previous work, efficiently recognizing the momentum of that work, in order to support the addition (reinstatement) of a species back toward a more realistic number. The Author has also clearly reviewed and referenced several previous workers, since Cutler’s (1994) reduction, who have found not only new species, but also that various groups that were lumped into a single species are most likely represented by a species complex. Yet, none of those workers pursued reinstatement as the Author has done here. The Reviewer has recognized this new study as significant, as it may stimulate others to follow using modern molecular genetic analyses to resolve sipunculan taxonomy. Therefore, this manuscript is timely, well-presented, validated and provocative . . . and it is clearly appropriate for meeting the Aims and Scope of PeerJ.

Annotated reviews are not available for download in order to protect the identity of reviewers who chose to remain anonymous.

·

Basic reporting

- English must be improve. Maybe, to be read for an English spoken person. Taxonomy has its own style and way to say things. So, sometimes ideas are not well written or lack of correct grammar.
- Literature is enough. Just one reference cited is lacking in the section.
- The document attached has many recommendations, changes, suggestions, questions, etc. that could improve the paper. Nevertheless, the document needs more than that. It needs more argumentation and to clarify many ideas.

The document is accepted with major changes and revisions. I will need to revise a new version.

Experimental design

See comments done in the document.

Validity of the findings

Document needs strong argumentation for being clearer in both morphological and molecular findings and analysis.
In the document, I have put several changes, suggestions, etc. in these sections.
As it is used in taxonomy, the discussion presented here could be divided in two. The first part must me fused to remarks (morphological analysis), and the second part must be fused to molecular analysis.
Thus, all the information will be more compacted and inclusive.
In molecular findings, the discussion and analysis are far to be clear. I recommend, if the molecular section does not improve, could be deleted since in the present state, does not give relevant information to the reinstatement of the species.
On the other hand, morphological argumentation is still weak. Author has to include information about the characters in other close species, as well as the discussion about that the characters chosen are very stable and informative.

Additional comments

The document needs a new version and new revision.
Please, check the English, but particularly, check the telegraphic style in the redescription.
If author does not attend some comments, they have to be explained why they were not included in the new version.
All comments are done in the word file.

---

## Round 0.2 · accepted · Accept

Dear Dr. Silva-Morales,

Many thanks for attending all reviewer's comments. I have made minor suggestions in the attached PDF. One thing that would be necessary to include is a statement that the voucher specimens of the sequenced individuals were not morphologically examined to confirm their generic placement.

Looking forward to seeing this manuscript published.
Best,
Wagner